# The Effects of Consuming Mineral Water from the “Topla Voda” Spring on the Body Composition and Functional and Biochemical Parameters of Professional Male Handball Athletes: A Pilot Study

**DOI:** 10.3390/sports13040100

**Published:** 2025-03-26

**Authors:** Djordje Batinic, Andrija Djuranovic, Milos Maletic, Sanja Stankovic, Vladimir Zivkovic, Dejan Stanojevic, Sergey Bolevich, Milan Savic, Vladimir Jakovljevic

**Affiliations:** 1Medical Department, Serbian Institute of Sports and Sports Medicine, 11030 Belgrade, Serbia; djbatinic87@gmail.com (D.B.); maleticmisa@gmail.com (M.M.); drvladakgbg@yahoo.com (V.J.); 2Department of Medical Biochemistry, Faculty of Medical Sciences, University of Kragujevac, 34000 Kragujevac, Serbia; sanjast2013@gmail.com; 3Center for Medical Biochemistry, University Clinical Center of Serbia, 11030 Belgrade, Serbia; 4Department of Physiology, Faculty of Medical Sciences, University of Kragujevac, 34000 Kragujevac, Serbia; vladimirziv@gmail.com; 5Center of Excellence for the Study of Redox Balance in Cardiovascular and Metabolic Disorders, University of Kragujevac, 34000 Kragujevac, Serbia; 6Department of Pharmacology, First Moscow State Medical University, 119435 Moscow, Russia; 7Special Hospital Merkur, 36210 Vrnjacka Banja, Serbia; dejan@vrnjcispa.rs; 8Department of Human Pathology, First Moscow State Medical University, 119435 Moscow, Russia; bolevich2011@yandex.ru

**Keywords:** hydration, athlete, mineral water, CPET, hemoglobin

## Abstract

Adequate hydration is crucial to an athlete’s health and performance. There is some evidence that the different compositions of various mineral water types may improve exercise performance and affect different biomarkers. The aim was to investigate the consumption of mineral water from the “Topla voda” spring in terms of its safety profile and its effect on body composition and functional and biochemical parameters in professional athletes. During the preparation phase of their mesocycle, 14 male professional handball players underwent a complete sports medical screening exam with a cardiopulmonary exercise test (CPET), blood gas analysis, and oxidative stress marker dynamics taken at four points during the CPET. The athletes were then randomized into two equal groups; the first group consumed mineral water, and the second group consumed tap water. After four weeks, the biochemical analysis and CPET were repeated. Routine analyses showed that the “mineral water” group had increased their mean corpuscular hemoglobin (ANCOVA = 0.050) and mean corpuscular hemoglobin concentration (ANCOVA = 0.001) and had a greater metabolic equivalent of task (MET) value at the end of the test (ANCOVA = 0.049), with no significant changes in the other measured parameters. Consuming “mineral water” appears to be safe, with some potential positive effects compared with tap water, mostly in terms of hemoglobin parameters and exercise tolerance.

## 1. Introduction

Water is the principal chemical constituent of the human body. For an average young adult male, their total body water represents 50% to 70% of their body weight [1]. Variability in total body water is primarily due to differences in body composition, and various techniques can measure hydration [2]. The net body water balance is regulated by thirst and hunger drives, coupled with ad libitum access to food and fluids that offset water losses. Among the greatest challenges to body water homeostasis are exercise and exercise–heat stress. Normal hydration can be achieved with a wide range of water intakes by both sedentary and active people across their lifespan [1]. Adequate fluid intake can be defined as the volume of fluid (from water, beverages, and food) sufficient to replace water losses and provide for solute excretion. A wide range of fluid intakes is compatible with normal hydration, and a person’s total body water varies narrowly from day to day by 600 to 900 mL (<1% of body mass) [3].

Rehydration during exercise should meet the primary goal of preventing the loss of more than 2% of a person’s body weight from water deficit to avoid performance deterioration and negative health outcomes [4,5]. The effects of hypohydration include reduced blood volume, especially in hot environments; increased skin blood flow and increased sweat rate; a subsequently increased core temperature, causing further cardiovascular strain; decreased venous return; preload; and a compensatory increased heart rate (HR) [6]. Considering the importance of specific ions for cell membrane stability and conductivity, especially in excitable tissues responsible for exercise and adequate physical exertion, not only the amount but also the composition of fluid has great importance during exercise. Sodium is the main electrolyte lost in sweat (20–70 mEq/L). Sodium supplementation during exercise is often required for heavy and “salty” sweaters to maintain their plasma volume and plasma sodium balance [7]. A precise refueling strategy during exercise should be taken into account when considering the type, duration, and level of exercise [8].

On the other hand, exercise-induced oxidative stress has been researched for a long time. It is still unclear whether the increases in reactive oxygen species are detrimental to health and performance [9]. As shown in studies investigating adaptation in altitude training, there is not enough evidence to recommend high-dose single-antioxidant supplementation, as this may actually impair endurance and altitude-based training adaptation. However, adding ample amounts of antioxidant-rich foods into athletes’ diets does not produce this detrimental effect [10]. To the authors’ knowledge, no investigation into the ingestion of various water types and their effects on redox parameters has been conducted.

There is some evidence that alkaline (hydrogen-rich) water, in some cases, improves exercise performance [11] and even affects blood pH in physically active men [12]. The ability to attenuate the rate of hydrogen ion (H^+^) accumulation in the muscles during exercise and/or enhance its removal from the muscles during recovery may affect the extent of exercise-induced disruption of excitation–contraction coupling, glycolytic flux, and phosphocreatine recovery and permit increased performance during continuous and intermittent high-intensity exercise [13]. In a study by Steffl et al. [14], where highly alkaline water was used for 3 consecutive days, a degree of aerobic performance improvement was shown. A recent study by Chiron et al. [15] investigated how bicarbonate-rich water affects various parameters when combined with either an “alkalizing” or “acidizing” diet. The results show that bicarbonate-rich water can alter the acid–base balance during warm-up and after high-intensity exercise, potentiating the possible beneficial effects of an alkalizing diet on the acid–base balance and reducing the acid load induced by an acidifying diet. No beneficial effect was observed regarding maximal exercise. When used as a supplement, sodium bicarbonate may increase the body’s extracellular buffering capacity, with potential beneficial effects on sustained high-intensity exercise performance [16].

Regarding all the factors mentioned as affecting exercise performance, the aim of our study was to investigate the consumption of mineral water from the “Topla voda” spring in terms of its safety profile and its effect, given its higher bicarbonate content, on body composition and functional and biochemical parameters in professional athletes.

## 2. Materials and Methods

### 2.1. Participants

The study involved 14 male professional handball players with an average age of 23.7 ± 4.9 years, all of whom had extensive experience in this sport. It was conducted during the preparation phase of their mesocycle.

### 2.2. Experiment Design

This was a prospective, randomized controlled study.

### 2.3. Study Protocol

The athletes underwent a comprehensive sports medical screening exam that included a physical exam, resting ECG (Cardiovit AT-102 G2, Schiller, Baar, Switzerland), body composition assessment, routine laboratory analyses, heart echocardiogram (CX50, Philips, Amsterdam, The Netherlands and Acuson Juniper, Siemens, Munich, Germany), and a cardiopulmonary exercise test (CPET). Additionally, fingertip blood gas analysis was performed, and oxidative stress markers were evaluated.

A complete examination was conducted during the initial visit. After at least a 3-h fasting period, a sample for the basic biochemical panel was collected while at rest. Given the dynamic changes during the test, blood gas analysis and oxidative stress parameters were sampled at 8 a.m. (basal) and at four time points through the CPET: just before the test began (point 1), during maximal exertion/end of the test (point 2), five minutes into the rest phase (point 3), and ten minutes into the rest phase (point 4).

Following the initial test, the athletes were divided into two groups by random allocation. The first group (*n* = 7) consumed mineral water, “Topla voda”, while the second group (*n* = 7) consumed tap water.

The training protocol during the four-week preparation period included handball training (8 h per week), strength training (6 h per week), conditioning (6 h per week), and three friendly games. During this time, each athlete received their own water bottle, and their water intake was carefully monitored. Water intake was regulated before, during, and after training sessions or friendly games. The average intake was 2.1 ± 0.5 L during this period. Assistant coaches or team physiotherapists provided the water and monitored the intake. The water was refrigerated at +4 °C.

After four weeks of consuming either mineral water or tap water, biochemical analyses and CPETs were repeated during the second (final) visit. Testing was conducted each day at the same time (10 a.m.), and athletes fasted for at least three hours before the test. Blood was collected from the cubital vein in properly frozen vacutainers until it was analyzed.

#### Water Characteristics

Table 1 presents the chemical and mineral properties of the mineral water from the “Topla voda” spring. The control group consumed tap water.

### 2.4. Body Composition

The bioelectrical impedance analysis (BIA) method (InBody 370, InBody, Seoul, Republic of Korea) was employed in order to obtain the following anthropometric parameters: body height, body weight, and body composition indicators (bone mass, soft tissue mass, total fat mass, skeletal muscle mass (SMM), fat percentage, total body water, and body mass index (BMI)).

### 2.5. Functional Parameters

The functional parameters examined during CPET (Quark CPET metabolic cart and h/p/cosmos pulsar treadmill, Cosmed, Rome, Italy) included absolute and relative maximum oxygen consumption (VO_2_max), respiratory exchange ratio (RER), maximal heart rate (HRmax), respiratory reserve, aerobic threshold, and anaerobic threshold. After the test, the modified Borg rating of perceived exertion (RPE) scale of 0–10 was used to assess the subjective feeling of maximal exertion. The metabolic equivalent of task (MET) value was automatically calculated by the metabolic cart’s automatic measurement system, which determines MET values based on weight, VO_2_, and test duration.

### 2.6. Routine Laboratory Analyses

The laboratory analyses conducted on the Mythic 18 analyzer (Orphee, Plan-les-Ouates, Switzerland) included a complete blood count of leukocytes (WBC), erythrocytes (RBC), platelets (PLT), lymphocytes (Lymph), granulocytes (Gran), hemoglobin (Hb), hematocrit (HCT), mean corpuscular volume (MCV), mean corpuscular hemoglobin (MCH), mean corpuscular hemoglobin concentration (MCHC), mean platelet volume (MPV), and plateletcrit (PCT). The biochemical parameters of sodium (Na), potassium (K), glucose, total cholesterol, LDL cholesterol, HDL cholesterol, triglycerides, urea, creatinine, AST, ALT, total proteins, iron, total bilirubin, direct bilirubin, and ferritin were measured in serum samples by using the Aries analyzer (Instrumentation Laboratory, Milano, Italy).

### 2.7. Blood Gas Analysis

The pH, blood gases, lactate, and oximetry in heparinized whole blood were measured by using the ABL90 FLEX automated analyzer (Radiometer Medical ApS, Copenhagen, Denmark). Four different measuring principles employed in the sensors of the ABL 90 FLEX PLUS analyzer were utilized: potentiometry (pH and carbon dioxide tension (pCO_2_)), amperometry (D-glucose concentration (cGlu) and L(+)-lactate concentration (cLac)), optical pO_2_ (pO_2_), and spectrophotometry (total hemoglobin concentration (ctHb), oxygen saturation (sO_2_), a fraction of oxyhemoglobin in total hemoglobin (FO_2_Hb), a fraction of carboxyhemoglobin in total hemoglobin (FCOHb), a fraction of deoxyhemoglobin in total hemoglobin (FHHb), and a fraction of methemoglobin in total hemoglobin (FMetHb)). Derived parameters that were calculated or estimated based on measured and keyed-in data included the anion gap, the concentration of total carbon dioxide in plasma (ctCO_2_(P)), the concentration of total carbon dioxide in whole blood (CO_2_ content (ctCO_2_(B)), the concentration of hydrogen carbonate (HCO_3_^−^), standard bicarbonate (SBC), actual base excess (ABE), standard base excess (SBE), the ABE of fully oxygenated blood, the partial pressure of oxygen at half saturation (50%) in blood (p50), the partial pressure of oxygen in alveolar air (pO_2_(A)), the ratio of partial pressure of oxygen in arterial blood and alveolar air (pO_2_(a/A)), the respiratory index (RI), and the difference in the partial pressure of oxygen between alveolar air and arterial blood (pO_2_(A-a)).

### 2.8. Evaluation of Systemic Redox State

The redox status was evaluated spectrophotometrically by using the UV-1800 (Shimadzu, Kyoto, Japan) by measuring the levels of prooxidative parameters, hydrogen peroxide (H_2_O_2_), superoxide anion radical (O_2_^−^), nitrites (NO_2_^−^), and the index of lipid peroxidation (TBARS) in plasma. The activities of the corresponding antioxidative enzymes superoxide dismutase (SOD), catalase (CAT), glutathione S-transferase(s) (GST(s)), glutathione peroxidase (GPx), and reduced glutathione (GSH) were measured in erythrocytes in the same manner.

### 2.9. Determination of Prooxidative Parameters

The degree of lipid peroxidation in plasma was assessed by measuring thiobarbituric acid reactive substances (TBARS) by using 0.4 mL of 1% thiobarbituric acid (TBA) in 0.05 mL of NaOH mixed with 0.8 mL of plasma, incubated at 100 °C for 15 min, and measured at 530 nm. Distilled water served as a blank probe. The TBA extract was prepared by combining 0.8 mL of plasma with 0.4 mL trichloroacetic acid (TCA). The samples were then chilled on ice for 10 min and centrifuged for 15 min at 6000 rpm [17]. Nitric oxide (NO) decomposes rapidly to produce the stable nitrite/nitrate metabolites. The method for detecting the plasma nitrite levels is based on the Griess reaction. Nitrites were quantified as an index of NO production by using the Griess reagent, which forms a purple diazocomplex [18]. A mix of 0.1 mL of 3 N perchloric acid (PCA), 0.4 mL of 20 mM ethylenediaminetetraacetic acid (EDTA), and 0.2 mL of plasma was placed on ice for 15 min and then centrifuged for 15 min at 6000 rpm. After removing the supernatant, 220 µL of K_2_CO_3_ was added. Nitrites were measured at 550 nm with distilled water as a blank probe. The level of O_2_^−^ was assessed by using a Nitro Blue Tetrazolium (NBT) reaction in TRIS buffer with plasma read at 550 nm. Distilled water was again used as a blank probe [19]. The determination of hydrogen peroxide (H_2_O_2_) concentration relies on the oxidation of phenol red solution (PRS) by using H_2_O_2_ through a reaction catalyzed by the enzyme peroxidase (POD) from horseradish [20]. Samples of 200 µL were combined with 800 µL of PRS and 10 µL of POD in a 1:20 ratio and measured at 610 nm.

### 2.10. Determination of Antioxidative Enzyme Activity

Isolated RBCs were washed three times with three volumes of ice-cold 0.9 mmol/L NaCl, and hemolysates containing approximately 50 g of Hb/L, prepared according to McCord and Fridovich [21], were used for determining catalase (CAT) activity. The measurement of CAT activity was conducted as per Beutler [22]. Lysates were diluted with distilled water (1:7 *v*/*v*) and treated with chloroform–ethanol (0.6:1 *v*/*v*) to eliminate hemoglobin. Next, 50 µL of CAT buffer, 100 µL of sample, and 1 mL of 10 mM H_2_O_2_ were added to the samples. Detection occurred at 360 nm with distilled water serving as a blank probe. SOD activity was determined by using the epinephrine method by Misra and Fridovich [23]. A mixture of 100 µL of lysate and 1 mL of carbonate buffer was prepared, followed by the addition of 100 µL of epinephrine. Detection was carried out at 470 nm. This method is classified as a “negative”-type group since it monitors the reduction in the rate of autoxidation in an alkaline medium, which is dependent on O_2_. The GSH concentration level was determined through GSH oxidation with 5.5-dithiobis-6.2-nitrobenzoic acid by utilizing the Beutler method [24]. Absorbance measurement was performed at the wavelength of maximum absorption at 420 nm.

### 2.11. Sample Size

The sample size calculation was based on previous research investigating the effect of mineral alkaline water consumption over three days in recreational athletes [14], utilizing G*Power software (version 3.1.9.7; Heinrich-Heine-Universität Düsseldorf, Düsseldorf, Germany) [25]. Drawing from the findings of the study by Steffl et al. [14], which included the hypothesis with the largest sample and the expected minimal difference between groups in the investigated parameters, the total number of participants was determined to be at least 12.

### 2.12. Statistical Analysis

Depending on the type of variables, data are presented as n (%) or means ± standard deviations. To test the differences in means between two independent samples (mineral and tap water), an independent samples t-test was used. A paired samples t-test was applied to examine differences in means between two repeated measurements (pre–post). An ANOVA for repeated measures was used to assess differences in multiple repeated measurements for each type of water separately. The Bonferroni procedure was applied for multiple comparisons of repeated measures data. To model the relationship between outcome numerical variables and the type of water, adjusted for baseline values, an ANCOVA was performed. A linear mixed model was used to model the relationship of dependent variables in repeated measurements concerning the type of water and measurement time. Statistical hypotheses were tested at an 0.05 (alpha) statistical significance level. All data were processed by using IBM SPSS Statistics 24 (IBM Corporation, Armonk, NY, USA) and R-4.0.0 software (The R Foundation for Statistical Computing, Vienna, Austria).

## 3. Results

### 3.1. Body Composition Parameters

As presented in Table 2, the two groups of athletes did not show significant differences in any body composition parameters, including blood pressure and HR at rest.

### 3.2. Biochemical Parameters

Table 3 presents the hematological and biochemical parameters. The concentration of Hb increased in both groups, nearly reaching statistical significance in the “mineral water” group. Two parameters related to Hb and iron metabolism that showed statistical significance in the “mineral water” group after four weeks are MCH (ANCOVA = 0.05) and MCHC (ANCOVA = 0.001). Other parameters did not show any statistically significant differences.

### 3.3. CPET Parameters

CPET parameters are presented in Table 4. Relative VO_2_ uptake was higher after four weeks in both groups. The increase was somewhat greater in the “mineral water” group, although it was not statistically significant. The athletes consuming “mineral water” demonstrated a higher MET value at the end of the test (ANCOVA = 0.049), and their increase was also greater after four weeks compared with the control group (*p* = 0.04). These athletes attained higher HRmax during the second visit at the end of the test compared with the control group (*p* = 0.04), even when expressed as a percentage of theoretical HRmax (*p* = 0.01). The RPE scale showed lower values in the “control” group during the first visit (*p* = 0.001), but RPE did not differ during the second visit. Other parameters did not show a statistically significant difference.

### 3.4. Blood Gas Analyses

The blood gas analyses are presented in Table 5. The statistical analyses compared parameters between the two groups and within each group (between the first and second measurements). The arterial and alveolar PaO_2_ (a/ApO_2_) and the gradient of PO_2_ between alveolar and arterial blood (AaDpO_2_) did not show statistical differences across the four measurements during each visit, nor did the comparison of the first and second visits between and within groups. The ABE exhibited negative values (indicating a buffer deficit, primarily due to bicarbonates). The ABE values were higher in both groups during the second visit. The anion gap increased in all three measurements compared with the others, with no significant difference between the groups. The bicarbonate ion concentration correlated with the dynamics of the anion gap. Other electrolytes did not show statistically significant differences across all measurements.

### 3.5. Oxidative Stress Markers

The oxidative stress markers and antioxidative defense results are presented in Table 6 and Table 7. The statistical analyses compared parameters between two groups and within each group (between the first and second measurements). The values for TBARS, NO_2_^−^, and O_2_^−^ exhibited no statistically significant difference across all measurements. The same trend was observed for H_2_O_2_. The activity of most enzymes involved in antioxidative protection was found to be higher in both groups during the second measurement. Although not statistically significant, increased activity of SOD, GSH, and GPx was noted during the second visit in the “mineral water” group.

## 4. Discussion

This pilot experiment aimed to investigate the effect of consuming “mineral water” on body composition and various biochemical and CPET parameters in professional male handball athletes when compared with athletes consuming tap water.

The data from BIA indicated no difference between the two groups during the first and second visits, allowing for a relevant comparison of these groups. The slight favorable changes in muscle mass and fat mass can be attributed to the appropriately periodized and planned training over this four-week period. Generally, each method for measuring body composition has its own strengths and weaknesses [26], but all athletes in this experiment were assessed under the same conditions during both visits. After four weeks of intensive training preparation, consuming mineral water did not negatively impact body composition. However, a longer consumption period is necessary to explore potential positive effects on these parameters.

The two groups generally showed no significant changes in basic biochemical panel results over the two visits. The mineral water group exhibited a slightly greater increase in Hb, although this was not statistically significant, likely due to the small sample size. However, MCV and MCHC demonstrated a statistically significant increase in the mineral water group compared with the control group. Given the hemoglobin’s role in oxygen transport, improved performance and recovery can be anticipated. Examining the differences in the training type, volume, and sport, there are no consistent findings regarding which athletes will experience the greatest increase in Hb parameters [27]. All athletes in this experiment started with optimal iron reserves, indicated by serum ferritin levels above 50 ng/mL, allowing for adequate erythropoiesis in response to training. No significant differences were observed in the serum ferritin levels during the second visit. The positive effect on Hb parameters is likely reflected during CPET, where athletes in the mineral water group achieved higher MET values. While athletes in the control group reported lower RPE during the first visit, there was no difference between groups during the second visit. The higher HRmax reached by the mineral water group suggests that these athletes adapted better to the training protocol, as a reduced HRmax can indicate “overtraining” or increased fatigue [28]. These observations imply that the group consuming mineral water responded more favorably to the training protocol or developed a higher fatigue threshold for high-intensity work compared with the control group. It is also important to note that all athletes underwent the same training protocol and ate the same standardized meals. To our knowledge, no study has examined the correlation among hydration, various water characteristics, and iron status in athletes. Recent data from our laboratory [29] were used to investigate the effects of low mineral content water in basketball players and found no improvement in Hb status. The recommended iron concentration for athletes [30] is unlikely to be present in any drinking water; however, through other mechanisms that positively influence homeostasis, athletes may benefit from consuming water with characteristics similar to those of mineral water. This hypothesis would need further investigation with more participants, ideally across different sports.

As mentioned in the introduction, a study by Chiron et al. [15] showed no beneficial effect, as participants consumed the water for only one week in a crossover design. No beneficial effect was observed concerning maximal exercise. In this study, where the water was consumed over four weeks, the authors hypothesize that the increased bicarbonate content of the “mineral water” could account for these results. Further studies with a closer monitoring of various biochemical parameters should be conducted to clarify this hypothesis.

Specific rehydration protocols involving various oral solutions containing different electrolytes and carbohydrates are recommended around training, particularly as a strategy during and after prolonged activity [31]. This was not the focus of our study, which examined the “habitual” water intake as a baseline, in addition to proper nutrition and rehydration strategies. As shown in a study [32] investigating muscle cramping, solely consuming mineral water is not an effective strategy following dehydration under hot conditions. Richard et al. [33] investigated the “acute” intake of different types of water around an exercise test, noting the lowest pH and muscular fatigue with bicarbonate water. Another study by Harris et al. [34] showed that mineral water with varying properties could improve rehydration after exercise, as indicated by serum osmolarity.

Capillary blood gas sampling from the fingertip was utilized to assess this study’s acid–base and ventilation status. Due to the sampling during CPET, performing arterial blood sampling was neither safe nor practical. For most blood gas parameters, the results do not differ significantly between these two methods [35]. The a/ApO_2_ nearly reached statistical significance, favoring the mineral water group, which may indicate better efficacy in O_2_ gas exchange across the alveolar membrane. However, as noted by Zavorsky et al. [35], arterial oxygen pressure can vary with this method, so this result should be interpreted with caution. ABE values were higher at the second visit, primarily due to the intense training protocol. Blood pH values were lower in the mineral water group, suggesting that four weeks was insufficient for the mineral water to demonstrate its buffering capacity, considering the anticipated effects of its characteristics, particularly bicarbonate concentration. An expected increase in the anion gap arose from metabolic acidosis resulting from the depletion of bicarbonate and its buffering role. As mentioned earlier, one of the main expected benefits of mineral water is its effect on ion dynamics, which could likely be observed over a longer period than what has been investigated. Hb plays a role in buffering pH changes in blood through the combined transport of carbon dioxide and hydrogen ions (H^+^) as bicarbonate ions [36]. The previously noted increase in Hb parameters may also contribute to improved exercise tolerance and a higher fatigue threshold.

Various oxidative stress biomarkers are presented in exercise studies, including oxidants, antioxidants, markers of oxidative damage, and measurements of redox balance [37]. Antioxidative parameters did not show significant differences between the two groups. Given the intense training protocol, the antioxidative enzyme activity was, as anticipated, increased during the second visit, with slightly higher activity observed in SOD, GSH, and GPx in the mineral water group. This result indicates that mineral water had a slight positive effect on antioxidative capacities, serving as the first and last line of defense against oxidative stress. However, a longer follow-up would be necessary for a valid conclusion with more participants. A similar trend was noticed in oxidative stress biomarkers, where only the H_2_O_2_ values were statistically significant compared with the control. This may serve as a foundation for further investigation, particularly regarding oxidative stress.

Based on this study, we can conclude that this type of mineral water did not significantly alter body composition. The appropriate training strategy for athletes can lead to positive outcomes regarding body composition, as was also observed with routine biochemical parameters. Conversely, markers of Hb metabolism exhibited more favorable results compared with the control group, potentially resulting in better training adaptation, enhanced performance, and improved recovery. Additionally, the consumption of “mineral water” led to better CPET parameters and their dynamics compared with the control group. This is supported by the values of relative VO_2_max and MET, indicating better exercise tolerance. The effects may become more pronounced if this water is consumed for an extended period (longer than four weeks).

Furthermore, blood gas parameters indicated greater metabolic acidosis in all athletes, suggesting inadequate bicarbonate production. Consuming mineral water can potentially benefit this ion dynamic and create a positive environment. Additionally, oxidative stress parameters did not show any adverse changes in the “mineral water” group; therefore, there was no increased oxidative damage. Slightly higher activity of antioxidative enzymes was not observed compared with the control group.

Four weeks was insufficient time for “mineral water” and its increased bicarbonate concentration to demonstrate any significant buffer activity, particularly in athletes with elevated blood lactate levels and lower blood pH values.

## 5. Conclusions

In general, during this phase of investigating the influence of mineral water on all mentioned parameters in professional male handball athletes, it can be concluded that consuming “mineral water” is safe, with some potential positive effects compared with tap water, primarily on Hb concentration parameters and exercise tolerance. All these effects may be more apparent if this water is consumed over a longer period of time (more than four weeks). Therefore, future research will aim to explore the effects of consuming “mineral water” for longer with a larger group of athletes included, along with additional biochemical parameters to further clarify the mechanisms of these effects. Upcoming prospective studies should focus on different types of water and various sports (aerobic, anaerobic, or mixed) and extend the follow-up period to include comparisons during the preparatory, competitive, and recovery phases of elite athletes.

### Limitations

The authors recognize the limitations of this study. The small sample size and brief duration of the experiment are insufficient for drawing solid conclusions. There are differences in body composition, specifically body weight, muscle mass, and fat-free mass. Although these differences are not statistically significant, this can be a consequence of a small sample size. Concerning nutrition, while the athletes were provided with standard meals during the preparation phase, the composition of these meals was not monitored. The sweat rate was not measured, and some athletes may have needed more water than was made available. This pilot study explores the effects of mineral water consumption, and the authors will address these limitations in future experiments.

## Figures and Tables

**Table 1 sports-13-00100-t001:** Mineral water properties.

Oxygen saturation (%)	33.4
Dry residue at 180 °C (mg/L)	1996
pH	6.7
Nitrates (mg/L)	<1.0
Nitrites (mg/L)	<0.005
Fluorides (mg/L)	3.45
Chlorides (mg/L)	38
Sulphates (mg/L)	8.6
Sulphites (mg/L)	0.022
Cyanides (mg/L)	<0.01
Bicarbonates (mg/L)	2135.0
Dissolved carbon dioxide (mg/L)	666
Phenols (mg/L)	<0.003
Water hardness (oN)	33.6
Sodium (mg/L)	532
Potassium (mg/L)	69.3
Calcium (mg/L)	66.1
Magnesium (mg/L)	56.3
Iron (mg/L)	0.75
Manganese (mg/L)	0.2
Copper (mg/L)	<0.03
Arsenic (mg/L)	<0.01
Barium (mg/L)	<0.05
Cadmium (mg/L)	<0.002
Lead (mg/L)	<0.005
Mercury (mg/L)	<0.0005
Selenium (mg/L)	<0.002
Antimony (mg/L)	<0.002
Chromium (mg/L)	<0.02

**Table 2 sports-13-00100-t002:** Subject characteristics.

Parameter	Measurement	Mineral Water	Tap Water	*p*-Value
Age		24 ± 6	23 ± 6	NS
Resting heart rate (beats per min)		65 ± 13	57 ± 9	NS
Resting systolic blood pressure (mmHg)		110 ± 12	111 ± 10	NS
Resting diastolic blood pressure (mmHg)		70 ± 9	65 ± 10	NS
Height (cm)		186 ± 6	192 ± 3	NS
Weight (kg)	Before	94.8 ± 9.6	99.1 ± 10.1	NS
	After	94.9 ± 9.1	99.8 ± 10.0	
BMI (kg/m^2^)	Before	27.3 ± 3.2	26.8 ± 2.3	NS
	After	27.4 ± 3.2	27.0 ± 2.3	
Body fat percentage (%)	Before	14.4 ± 4.6	14.3 ± 4.2	NS
	After	14.3 ± 4.6	13.7 ± 3.7	
Muscle mass (kg)	Before	46.8 ± 3.8	49.9 ± 4.9	NS
	After	46.9 ± 3.7	49.0 ± 5.5	
Muscle mass percentage (%)	Before	49.5 ± 2.5	50.3 ± 2.7	NS
	After	49.6 ± 2.4	49.7 ± 2.4	
Total body water (kg)	Before	59.2 ± 4.6	63.5 ± 6.3	NS
	After	59.3 ± 4.3	62.9 ± 6.3	
Free fat mass (kg)	Before	80.9 ± 6.3	86.9 ± 8.5	NS
	After	81.1 ± 5.9	86.0 ± 8.6	

NS: not significant; BMI: body mass index. Results are expressed as means ± standard deviations.

**Table 3 sports-13-00100-t003:** Routine laboratory analyses.

Parameter	Measurement	Mineral Water	Tap Water	*p*-Value	ANCOVA
Leukocytes (×10^9^/L)	Before	5.4 ± 1.1	6.0 ± 0.6	NS	NS
	After	5.4 ± 0.8	5.8 ± 1.8	NS	
Lymphocytes (%)	Before	33.8 ± 10.6	32.7 ± 8.9	NS	NS
	After	33.6 ± 8.8	36.9 ± 7.6	NS	
Monocytes (%)	Before	5.0 ± 0.9	5.9 ± 0.9	NS	NS
	After	5.4 ± 1.1	6.3 ± 1.0	NS	
Granulocytes (%)	Before	61.2 ± 10.5	61.3 ± 8.6	NS	NS
	After	61.1 ± 9.4	56.7 ± 7.9	NS	
Erythrocytes (×10^12^/L)	Before	5.0 ± 0.3	4.7 ± 0.2	NS	NS
	After	5.1 ± 0.2	4.9 ± 0.1	NS	
Hemoglobin (g/L)	Before	143 ± 5	135 ± 9	NS	NS
	After	150 ± 9	142 ± 5	NS	
Hematocrit (l/L)	Before	0.43 ± 0.01	0.41 ± 0.03	NS	NS
	After	0.44 ± 0.02	0.43 ± 0.02	NS	
Mean corpuscular volume, MCV (fL)	Before	85.6 ± 3.7	86.2 ± 1.7	NS	NS
	After	89.0 ± 5.5	88.4 ± 1.8	NS	
Mean corpuscular hemoglobin, MCH (pg)	Before	28.6 ± 1.5	28.6 ± 0.9	NS	0.050
	After	29.7 ± 1.5	29.1 ± 0.5	NS	
Mean corpuscular hemoglobin concentration, MCHC (g/L)	Before	334 ± 7	332 ± 8	NS	0.001
	After	339 ± 4	330 ± 3	0.001	
Thrombocytes (×10^9^/L)	Before	250 ± 25	234 ± 43	NS	NS
	After	242 ± 31	244 ± 36	NS	
Sedimentation rate, SE (mm/h)	Before	2.6 ± 0.8	2.6 ± 0.9	NS	NS
	After	2.7 ± 0.8	4.7 ± 3.5	NS	
Potassium (mmol/L)	Before	4.3 ± 0.3	4.1 ± 0.3	NS	NS
	After	4.3 ± 0.3	4.4 ± 0.3	NS	
Sodium (mmol/L)	Before	142 ± 3	144 ± 1	NS	NS
	After	142 ± 2	141 ± 2	NS	
Glucose (mmol/L)	Before	4.9 ± 0.4	4.9 ± 0.3	NS	NS
	After	5.2 ± 0.2	5.3 ± 0.1	NS	
Total cholesterol (mmol/L)	Before	4.6 ± 0.5	4.1 ± 1.3	NS	NS
	After	4.4 ± 0.3	4.2 ± 0.9	NS	
HDL (mmol/L)	Before	1.6 ± 0.2	1.6 ± 0.2	NS	NS
	After	1.5 ± 0.2	1.5 ± 0.2	NS	
LDL (mmol/L)	Before	2.8 ± 0.6	2.4 ± 1.1	NS	NS
	After	2.5 ± 0.3	2.4 ± 0.8	NS	
Triglycerides (mmol/L)	Before	0.5 ± 0.1	0.5 ± 0.3	NS	NS
	After	0.7 ± 0.2	0.7 ± 0.4	NS	
Urea/Blood urea nitrogen (mmol/L)	Before	8.2 ± 1.7	7.7 ± 2.1	NS	NS
	After	6.8 ± 1.3	6.8 ± 1.6	NS	
Creatinine (μmol/L)	Before	106 ± 8	101 ± 11	NS	NS
	After	96 ± 6	88 ± 9	NS	
AST (U/L)	Before	47.7 ± 29.9	37.4 ± 16.9	NS	NS
	After	26.7 ± 6.2	30.1 ± 4.1	NS	
ALT (U/L)	Before	35.1 ± 18.8	26.3 ± 2.4	NS	NS
	After	24.9 ± 7.6	20.7 ± 4.1	NS	
Proteins (g/L)	Before	70.9 ± 2.7	73.1 ± 2.4	NS	NS
	After	67.9 ± 3.7	69.1 ± 2.5	NS	
Iron (μmol/L)	Before	23.5 ± 6.7	21.9 ± 6.9	NS	NS
	After	14.8 ± 6.2	17.4 ± 6.8	NS	
Total bilirubin (μmol/L)	Before	17.6 ± 3.4	15.9 ± 5.5	NS	NS
	After	13.5 ± 4.9	12.0 ± 4.8	NS	
Direct bilirubin (μmol/L)	Before	3.6 ± 0.5	3.4 ± 0.9	NS	NS
	After	3.0 ± 1.2	2.7 ± 0.7	NS	
Ferritin (ng/mL)	Before	125 ± 54	84 ± 21	NS	NS
	After	108 ± 70	73 ± 16	NS	

NS: not significant. Results are expressed as means ± standard deviations.

**Table 4 sports-13-00100-t004:** CPET parameters.

Parameter	Measurement	Mineral Water	Tap Water	*p*-Value	ANCOVA
Maximal speed (km/h)	Before	11.7 ± 0.5	10.6 ± 0.8	0.008	NS
	After	11.4 ± 0.5	10.9 ± 0.9	NS	
Maximal incline (%)	Before	12.0 ± 0.0	10.9 ± 1.1	0.030	NS
	After	12.0 ± 0.0	11.1 ± 1.1	NS	
Test duration (s)	Before	601 ± 58	542 ± 62	NS	NS
	After	580 ± 31	566 ± 85	NS	
MET	Before	14.9 ± 0.9	14.5 ± 1.3	NS	0.049
	After	16.0 ± 1.0	14.6 ± 1.2	0.040	
HRmax (beats per min)	Before	189 ± 5	180 ± 11	NS	NS
	After	186 ± 4	175 ± 10	0.030	
Predicted/theoretical HRmax (%)	Before	97 ± 3	92 ± 4	0.024	NS
	After	95 ± 4	89 ± 3	0.010	
Heart rate recovery 1st minute	Before	165 ± 9	152 ± 8	0.013	NS
	After	163 ± 7	152 ± 12	NS	
Heart rate recovery 3rd minute	Before	104 ± 12	98 ± 13	NS	NS
	After	112 ± 14	101 ± 8	NS	
Maximal systolic blood pressure (mmHg)	Before	194 ± 17	183 ± 14	NS	NS
	After	184 ± 19	187 ± 21	NS	
Maximal diastolic blood pressure (mmHg)	Before	66 ± 18	51 ± 7	NS	NS
	After	54 ± 5	49 ± 7	NS	
Ventilatory anaerobic threshold, VAT (HR)	Before	184 ± 6	176 ± 10	NS	NS
	After	178 ± 4	168 ± 9	0.031	
Aerobic threshold, AT (HR)	Before	173 ± 9	162 ± 10	NS	NS
	After	165 ± 4	157 ± 9	NS	
Respiratory exchange ratio, RER	Before	1.02 ± 0.01	1.03 ± 0.02	NS	NS
	After	1.08 ± 0.06	1.05 ± 0.05	NS	
Maximal VO_2_ uptake, VO_2_max (mL/kg/min)	Before	53.7 ± 2.8	51.9 ± 4.8	NS	NS
	After	55.8 ± 2.6	52.6 ± 4.5	NS	
Rating of perceived exertion, RPE	Before	8.7 ± 0.9	6.7 ± 0.9	0.001	NS
	After	8.1 ± 0.9	8.1 ± 1.1	NS	

NS: not significant. Results are expressed as means ± standard deviations.

**Table 5 sports-13-00100-t005:** Blood gas analysis results.

Parameter	Measurement		Mineral Water	Tap Water	*p*-Value	Mixed Effect Interaction
pO_2_(a/A) (%)	Before	1.	68.8 ± 3.9	68.9 ± 6.3	NS	NS
		2.	95.0 ± 20.5	88.8 ± 7.5	NS	
		3.	91.5 ± 5.4	83.8 ± 7.7	NS	
		4.	86.4 ± 5.6	79.9 ± 4.6	0.035	
	After	1.	69.9 ± 6.1	72.2 ± 3.9	NS	
		2.	88.9 ± 4.1	90.5 ± 8.9	NS	
		3.	89.6 ± 6.4	90.5 ± 1.8	NS	
		4.	85.5 ± 6.7	85.2 ± 3.9	NS	
ABE (mmol/L)	Before	1.	−0.03 ± 1.21	0.23 ± 0.45	NS	NS
		2.	−13.50 ± 2.82	−9.13 ± 1.73	0.006	
		3.	−14.81 ± 2.70	−9.03 ± 3.02	0.003	
		4.	−13.10 ± 2.87	−6.51 ± 2.67	0.001	
	After	1.	0.36 ± 1.20	0.30 ± 1.04	NS	
		2.	−15.21 ± 3.01	−11.57 ± 2.35	0.028	
		3.	−16.81 ± 4.03	−11.93 ± 2.53	0.022	
		4.	−14.79 ± 3.81	−9.76 ± 2.57	0.015	
Anion gap (mmol/L)	Before	1.	10.9 ± 1.9	10.8 ± 1.4	NS	NS
		2.	21.5 ± 2.6	18.5 ± 1.4	0.027	
		3.	23.5 ± 2.2	18.5 ± 3.3	0.007	
		4.	21.5 ± 2.9	16.7 ± 2.7	0.008	
	After	1.	10.1 ± 1.6	9.6 ± 1.1	NS	
		2.	22.4 ± 2.9	18.4 ± 1.7	0.010	
		3.	23.6 ± 3.5	19.8 ± 2.7	0.044	
		4.	22.3 ± 2.8	18.4 ± 2.4	0.015	
HCO_3_^−^ (mmol/L)	Before	1.	25.2 ± 1.5	25.7 ± 0.9	NS	NS
		2.	13.9 ± 2.2	17.5 ± 1.2	0.005	
		3.	12.2 ± 2.1	17.2 ± 2.4	0.002	
		4.	13.3 ± 2.7	18.8 ± 2.1	0.001	
	After	1.	25.3 ± 1.2	25.5 ± 0.7	NS	
		2.	14.1 ± 2.6	16.9 ± 1.5	0.034	
		3.	11.9 ± 1.9	15.7 ± 1.9	0.017	
		4.	12.8 ± 2.9	16.8 ± 1.9	0.011	

NS: not significant; pO_2_(a/A)**:** the ratio of the partial pressure of oxygen in arterial blood and alveolar air; ABE: actual base excess; HCO_3_^−^: the concentration of hydrogen carbonate. The samples were collected at four time points during the CPET: just before the test began (1.), at maximal exertion/end of the test (2.), five minutes into the rest phase (3.), and ten minutes into the rest phase (4.). Results are presented as means ± standard deviations.

**Table 6 sports-13-00100-t006:** The dynamics of oxidative stress parameters.

Parameter	Measurement		Mineral Water	Tap Water	*p*-Value	Mixed Effect Interaction
TBARS (µmol/mL)	Before	1.	0.99 ± 0.23	0.96 ± 0.24	NS	NS
		2.	1.02 ± 0.34	0.98 ± 0.26	NS	
		3.	1.04 ± 0.32	0.96 ± 0.25	NS	
		4.	0.98 ± 0.22	0.98 ± 0.23	NS	
	After	1.	1.18 ± 0.05	1.14 ± 0.16	NS	
		2.	1.13 ± 0.15	1.17 ± 0.14	NS	
		3.	1.11 ± 0.20	1.18 ± 0.14	NS	
		4.	1.16 ± 0.17	1.17 ± 0.13	NS	
NO_2_^−^ (nmol/mL)	Before	1.	4.25 ± 0.41	4.29 ± 0.38	NS	NS
		2.	4.48 ± 0.58	4.24 ± 0.39	NS	
		3.	4.09 ± 1.23	4.24 ± 0.33	NS	
		4.	4.33 ± 0.38	4.16 ± 0.41	NS	
	After	1.	5.66 ± 0.76	5.51 ± 0.58	NS	
		2.	5.88 ± 0.88	5.52 ± 0.55	NS	
		3.	5.80 ± 0.80	5.58 ± 0.59	NS	
		4.	5.61 ± 0.76	5.84 ± 0.64	NS	
O_2_^−^ (nmol/mL)	Before	1.	1.51 ± 0.46	3.30 ± 1.34	0.012	NS
		2.	3.39 ± 1.37	2.64 ± 1.29	NS	
		3.	1.70 ± 0.72	1.16 ± 0.58	NS	
		4.	2.64 ± 1.36	1.98 ± 1.59	NS	
	After	1.	1.13 ± 0.42	1.08 ± 0.59	NS	
		2.	1.46 ± 0.57	1.74 ± 1.04	NS	
		3.	3.77 ± 1.04	3.34 ± 1.01	NS	
		4.	2.54 ± 0.65	3.06 ± 0.78	NS	
H_2_O_2_ (nmol/mL)	Before	1.	2.59 ± 0.37	2.05 ± 0.54	NS	0.008
		2.	2.48 ± 0.60	2.14 ± 0.44	NS	
		3.	2.03 ± 0.30	1.99 ± 0.39	NS	
		4.	2.17 ± 0.45	2.15 ± 0.39	NS	
	After	1.	3.59 ± 0.24	3.83 ± 0.49	NS	
		2.	3.60 ± 0.39	4.00 ± 0.24	NS	
		3.	3.80 ± 0.50	3.86 ± 0.39	NS	
		4.	3.48 ± 0.33	3.78 ± 0.45	NS	

NS: not significant. TBARS: the index of lipid peroxidation measuring thiobarbituric acid reactive substances; NO_2_^−^: nitrites; O_2_^−^: superoxide anion radical; H_2_O_2_: hydrogen peroxide. The samples were collected at four time points during the CPET: just before the test began (1.), at maximal exertion/end of the test (2.), five minutes into the rest phase (3.), and ten minutes into the rest phase (4.). Results are presented as mean ± standard deviation.

**Table 7 sports-13-00100-t007:** The dynamics of antioxidant defense.

Parameter	Measurement		Mineral Water	Tap Water	*p*-Value	Mixed Effect Interaction
SOD (U/g Hg × 10^3^)	Before	1.	16.3 ± 8.1	18.6 ± 13.1	NS	NS
		2.	9.3 ± 3.1	16.3 ± 9.4	NS	
		3.	17.4 ± 7.3	16.3 ± 7.3	NS	
		4.	14 ± 7.7	13.9 ± 6.2	NS	
	After	1.	23.3 ± 18.5	25.6 ± 5.6	NS	
		2.	34.9 ± 13.9	24.4 ± 16.9	NS	
		3.	20.9 ± 9.2	22.1 ± 10.2	NS	
		4.	26.8 ± 18.0	26.8 ± 12.2	NS	
CAT (U/g Hg × 10^3^)	Before	1.	7.04 ± 3.76	3.84 ± 2.41	NS	NS
		2.	5.11 ± 3.92	6.82 ± 5.14	NS	
		3.	3.71 ± 3.78	5.33 ± 3.11	NS	
		4.	4.64 ± 2.79	4.25 ± 3.62	NS	
	After	1.	2.14 ± 1.60	3.57 ± 2.31	NS	
		2.	1.75 ± 1.63	3.04 ± 2.67	NS	
		3.	2.54 ± 2.38	3.07 ± 2.00	NS	
		4.	3.00 ± 1.62	3.54 ± 1.98	NS	
GSH (nmol/mL RBC × 10^3^)	Before	1.	68,448 ± 4682	68,726 ± 4376	NS	NS
		2.	76,609 ± 8794	72,834 ± 4915	NS	
		3.	78,718 ± 11,797	77,849 ± 8423	NS	
		4.	77,553 ± 4641	82,160 ± 9861	NS	
	After	1.	100,702 ± 16,809	91,875 ± 12,861	NS	
		2.	105,032 ± 105,032	91,875 ± 91,875	NS	
		3.	11,468 ± 11,468	11,527 ± 11,527	NS	
		4.	102,700 ± 102,700	92,763 ± 92,763	NS	
GPx (nmol/min/mL)	Before	1.	24.0 ± 23.7	39.6 ± 20.5	NS	NS
		2.	35.9 ± 13.0	31.1 ± 9.5	NS	
		3.	22.2 ± 18.6	39.9 ± 17.7	NS	
		4.	22.4 ± 19.8	24.7 ± 20.4	NS	
	After	1.	30.6 ± 14.7	31.5 ± 15.7	NS	
		2.	28.7 ± 18.9	32.6 ± 12.1	NS	
		3.	38.9 ± 14.1	23.9 ± 12.5	NS	
		4.	27.7 ± 13.2	42.4 ± 12.8	NS	
GST (mmol/mL/min)	Before	1.	1.94 ± 0.89	2.46 ± 0.65	NS	NS
		2.	2.69 ± 1.04	1.86 ± 1.11	NS	
		3.	2.03 ± 0.53	2.28 ± 0.52	NS	
		4.	2.06 ± 0.98	1.64 ± 0.93	NS	
	After	1.	2.23 ± 0.22	2.20 ± 0.74	NS	
		2.	2.05 ± 0.57	2.31 ± 0.81	NS	
		3.	2.22 ± 0.81	2.51 ± 0.77	NS	
		4.	2.15 ± 1.68	1.91 ± 1.43	NS	

NS: not significant. SOD: the activity of superoxide dismutase; CAT: the activity of catalase; GSH: reduced glutathione; GPx: glutathione peroxidase; GST: the activity of glutathione S-transferase. The samples were collected at four time points during the CPET: just before the test began (1.), at maximal exertion/end of the test (2.), five minutes into the rest phase (3.), and ten minutes into the rest phase (4.). Results are presented as mean ± standard deviation.

## Data Availability

The original contributions presented in this study are included in the article. Further inquiries can be directed to the corresponding author.

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
