# Peer review of "The Effects of Consuming Mineral Water from the “Topla Voda” Spring on the Body Composition and Functional and Biochemical Parameters of Professional Male Handball Athletes: A Pilot Study"

_sports, 2025, doi:10.3390/sports13040100_

Round 1
Reviewer 1 Report
Comments and Suggestions for Authors
In the abstract, please provide info that supports the conclusion as there is no info on fatigue perception for example. In fact, no data is presented in the results section on fatigue perception. What measurement was used for fatigue perception as the Borg scale is not really a scale for fatigue perception. Please clarify/revise.
L63. Note that the 2% of change in body mass by dehydration is based on research using furosemide, see e.g. doi:10.1152/japplphysiol.00367.2010. Dehydration by the drug or exercise has different effects on changes in plasma volume.
L103. I suggest to replace “23,7±4,9” with “23.7±4.9”.
Table 1. Change “Potassium(mg/L)” to “Potassium (mg/L)”
Table 1. I suggest to change the “,” in values with “.”.
L128. Was this allocation? If not, how was it randomized.
L147. Please revise “anaerobic threshold, and anaerobic threshold”.
Ls 156 and 161. Was glucose measured with two methods? Please clarify/revise. Similar for sodium and potassium.
L165. Please define abbreviations on first use, e.g. ctHb.
L211. Please revise “100gl ly- sate”.
L236. Please provide blood pressure and heart rate values in Table 2 and without decimal places.
Table 2. Please express mean and SD values of age and height without decimal places. In addition, please consider whether you need two decimal places for mean and SD values for the other parameters.
Table 2. Change “muscle” to “muscle”.
Table 3. hematocrit after mean value is missing for tap water.
Table 3, 5 and 7. Please consider whether two decimal places are required for the mean and SD values of parameters.
Table 4. Please express at least heart rate without decimal places and consider whether whether two decimal places are required for the mean and SD values of parameters.
Was there a measurement of VO2 in rest to quantify MET at exhaustion or was 3.5 ml (kg min) assumed to apply to all particpants. Please clarify.
In abbreviations, define TBARS.
Tables 5 to 7. I suggest to clarify in the legend what the numbers stand for for the measurements.
L375. There is mention of intense training. I suggest to provide some info what training was done during the 4 week period.
L413. What was the sample size required? Please provide a comment.
Author Response
Thank you for all the comments and suggestions. All changes are marked in red.
Comment 1: In the abstract, please provide info that supports the conclusion as there is no info on fatigue perception for example. In fact, no data is presented in the results section on fatigue perception. What measurement was used for fatigue perception as the Borg scale is not really a scale for fatigue perception. Please clarify/revise.
Response 1: Thank you for your comment. To avoid any possible misunderstandings and ensure an objective presentation of some parameters, we excluded fatigue from the abstract, keywords, and paper. Please see L45, L413, L469.
Comment 2: L63. Note that the 2% of change in body mass by dehydration is based on research using furosemide, see e.g. doi:10.1152/japplphysiol.00367.2010. Dehydration by the drug or exercise has different effects on changes in plasma volume.
Response 2: Thank you for your comment. We added the paper mentioned so the readers have additional information. The paper we originally cited is a practical recommendation by ACSM. Please see L64 and Ref 4 (in addition to the ACSM recomendations in Ref 5).
Comment 3: L103. I suggest to replace “23,7±4,9” with “23.7±4.9”.
Response 3: Corrected.
Comment 4: Table 1. Change “Potassium(mg/L)” to “Potassium (mg/L)”
Response 4: Corrected.
Comment 5: Table 1. I suggest to change the “,” in values with “.”.
Response 5: Corrected. All “,” are changed to “.”
Comment 6: L128. Was this allocation? If not, how was it randomized.
Response 6: It was random allocation. Corrected (L124)
Comment 7: L147. Please revise “anaerobic threshold, and anaerobic threshold”.
Response 7: Corrected (L154)
Comment 8: Ls 156 and 161. Was glucose measured with two methods? Please clarify/revise. Similar for sodium and potassium.
Response 8: Glucose and electrolytes were measured from the venous blood as a part of the routine laboratory analyses. Sorry for not clarifying. Corrected (L172).
Comment 9: L165. Please define abbreviations on first use, e.g. ctHb.
Response 9: Corrected (L172-180).
Comment 10: L211. Please revise “100gl ly- sate”.
Response 10: Corrected (L226)
Comment 11: L236. Please provide blood pressure and heart rate values in Table 2 and without decimal places.
Response 11: Corrected in Table 2
Comment 12: Table 2. Please express mean and SD values of age and height without decimal places. In addition, please consider whether you need two decimal places for mean and SD values for the other parameters.
Response 12: Corrected. Apart from the mentioned parameters (age, height, HR, blood pressure), hematocrit, RER, RPE in all of the tables, we expressed numbers with two digits (tens) with one decimal place and no decimal places where the number has three digits (hundreds). Numbers with one digit in oxidative stress parameters and antioxidant defense are expressed with two decimal places.
Comment 13: Table 2. Change “muscle” to “muscle”.
Response 13: Corrected.
Comment 14: Table 3. hematocrit after mean value is missing for tap water.
Response 14: Corrected. The mean value was missing.
Comment 15: Table 3, 5 and 7. Please consider whether two decimal places are required for the mean and SD values of parameters.
Response 15: Corrected. Apart from the mentioned parameters (age, height, HR, blood pressure), hematocrit, RER, RPE in all of the tables, we expressed numbers with two digits (tens) with one decimal place and no decimal places where the number has three digits (hundreds). Numbers with one digit in oxidative stress parameters and antioxidant defense are expressed with two decimal places.
Comment 16: Table 4. Please express at least heart rate without decimal places and consider whether whether two decimal places are required for the mean and SD values of parameters.
Response 16: Corrected. Apart from the mentioned parameters (age, height, HR, blood pressure), hematocrit, RER, RPE in all of the tables, we expressed numbers with two digits (tens) with one decimal place and no decimal places where the number has three digits (hundreds). Numbers with one digit in oxidative stress parameters and antioxidant defense are expressed with two decimal places.
Comment 17: Was there a measurement of VO2 in rest to quantify MET at exhaustion or was 3.5 ml (kg min) assumed to apply to all particpants. Please clarify.
Response 17: As stated, all participants performed CPET on Quark CPET metabolic cart and h/p/cosmos pulsar treadmill (Cosmed, Italy). We used the device's automatic measurement, where the program calculates METs based on the weight, VO2, and test duration. Explanation was added in L156.
Comment 18: In abbreviations, define TBARS.
Response 18: Corrected.
Comment 19: Tables 5 to 7. I suggest to clarify in the legend what the numbers stand for for the measurements.
Response 19: Corrected in Tables 5,6,7
Comment 20: L375. There is mention of intense training. I suggest to provide some info what training was done during the 4 week period.
Response 20: The explanation is added in the Study protocol. Please see L127.
Comment 21: L413. What was the sample size required? Please provide a comment.
Response 21: Corrected. Please se additonal section (L234)
Reviewer 2 Report
Comments and Suggestions for Authors
Authors present an interesting aspect of sport nutrition; the protocol is well organized nad conducted, anyway there are some possible improvement:
- It should be purposede a mechanism by which the minera water improve the final results: mineral salts content? Or more likely carbonated ions? It should be interested to have a comparison with different type of mineral water
- Using BIA it could be usefull to have a mesurement before and after training to check if there is an acute regulation
- Having used the CPET test it could be interesting report RF too that is linked to the percieved effert (see 10.3390/sports12100277)
- Thera are difference in weight, hieght and muscle mass, that could be noticed as possible bias
- The device choosen for BIA is not so reiliabel for Phase Angle that could be an usefull indicator for performance evaluation
- Conclusione should be improved, addressing possible new and more complete studies in the future
Author Response
Thank you for all the comments and suggestions. All changes are marked in red.
Comment 1: It should be purposede a mechanism by which the minera water improve the final results: mineral salts content? Or more likely carbonated ions? It should be interested to have a comparison with different type of mineral water
Response 1: Based on this study and earlier studies, the concentration of bicarbonated ions is probably the reason for the final results. Also, as stated in the conclusion, further research is needed to confirm this mechanism. Please see L401 and L473.
Comment 2: Using BIA it could be usefull to have a mesurement before and after training to check if there is an acute regulation
Response 2: We agree, but unfortunately, we did not have BIA available every day during training, only for the two tests 4 weeks apart.
Comment 3: Having used the CPET test it could be interesting report RF too that is linked to the percieved effert (see 10.3390/sports12100277)
Response 3: Thank you for the comment. Due to the large number of various parameters analyzed, we opted out of analyzing ventilatory response. We conclude that reporting solely RF without other parameters (EQO2, EQCO2, PETO2, PETCO2, etc.) would not paint the complete picture regarding ventilatory efficiency related to perceived effort. However, we want to analyze the CPET respiratory parameters separately and compare athletes from various sports.
Comment 4:Thera are difference in weight, hieght and muscle mass, that could be noticed as possible bias
Response 4: Thank you for this comment. We generally agree regarding these parameters, but in our study, there was no statistically significant difference in all anthropometric measurements between the two groups (Table 2); therefore, we would not mention this as a possible bias.
Comment 5: The device choosen for BIA is not so reiliabel for Phase Angle that could be an usefull indicator for performance evaluation
Response 5: Thank you for this comment. We acknowledge the strengths and weaknesses of BIA (please see L366) and any other available body composition measurement method. All of the participants in this experiment were measured under the same conditions during both visits.
Comment 6: Conclusione should be improved, addressing possible new and more complete studies in the future
Response 6: We expanded the conclusion. Please see L465 for further clarification.
Reviewer 3 Report
Comments and Suggestions for Authors
Introduction
The introduction is very appropriate. What is missing is an explanation of what special characteristics Toplavoda water has to make you want to do research with it. Just to elaborate a bit on the last paragraph on this issue and to stimulate the reader in the justification of the study
The aspect of iron deficiency and fatigue is somewhat more complicated than what is presented. In any case it is acceptable, although in male handball players this iron deficiency is not common and can occur in very specific subjects. Therefore, given that the model of volunteers for the study is not the appropriate one to observe this deficiency due to the practice of continuous exercise, and that subjects with iron deficiency or iron deficiency anaemia are not used, I do not see much need for this explanation.
Tables
In order to make it easier for the reader to focus on the process of the study, I advise to arrange the table of the composition of the water under study after the Study Protocol, and I would add logistical aspects, such as who provides the water, where and how it is stored so that it maintains its properties, etc. In short, dressing up Toplavoda's marketing.
In order to make the statistics easier to read, in all tables, decimals should be removed when the number has three digits (hundreds) and only one decimal place if the number has two digits (tens).
Regarding the P value and ANCOVA, when it is not significant I don't think it is necessary to state the number, with the acronym NS (not significant) is enough. Remember to explain what NS is in the table footer.
Discussion
The discussion is well developed within the limitations that are so well stated at the end of the discussion. Nothing to comment.
Conclusion
A Conclusion section is missing.
Author Response
Thank you for all the comments and suggestions. All changes are marked in red.
Introduction
Comment 1: The introduction is very appropriate. What is missing is an explanation of what special characteristics Toplavoda water has to make you want to do research with it. Just to elaborate a bit on the last paragraph on this issue and to stimulate the reader in the justification of the study.
Response 1: Thank you for the comment. We mentioned the paper by Chiron et al. (10.5114/jhk/182986) in the discussion, but we agree that, to clarify, we should first have mentioned it in the introduction section. Given the higher bicarbonate content, we thought about sodium bicarbonate as an ergogenic supplement and the potential mechanism as an extracellular buffer (added the IOC consensus statement, 10.1123/ijsnem.2018-0020 under new Ref 16). Please see L90 and L398.
Comment 2: The aspect of iron deficiency and fatigue is somewhat more complicated than what is presented. In any case it is acceptable, although in male handball players this iron deficiency is not common and can occur in very specific subjects. Therefore, given that the model of volunteers for the study is not the appropriate one to observe this deficiency due to the practice of continuous exercise, and that subjects with iron deficiency or iron deficiency anaemia are not used, I do not see much need for this explanation.
Response 2: We agree; therefore, we deleted this section.
Tables
Comment 3: In order to make it easier for the reader to focus on the process of the study, I advise to arrange the table of the composition of the water under study after the Study Protocol, and I would add logistical aspects, such as who provides the water, where and how it is stored so that it maintains its properties, etc. In short, dressing up Toplavoda's marketing.
Response 3: Table 1 position was altered. Please see L139.
We added the explanation. Please see L132.
Comment 4: In order to make the statistics easier to read, in all tables, decimals should be removed when the number has three digits (hundreds) and only one decimal place if the number has two digits (tens).
Response 4: Corrected. Apart from the mentioned parameters (age, height, HR, blood pressure), hematocrit, RER, RPE in all of the tables, we expressed numbers with two digits (tens) with one decimal place and no decimal places where the number has three digits (hundreds). Numbers with one digit in oxidative stress parameters and antioxidant defense are expressed with two decimal places.
Comment 5: Regarding the P value and ANCOVA, when it is not significant I don't think it is necessary to state the number, with the acronym NS (not significant) is enough. Remember to explain what NS is in the table footer.
Response 5: Corrected in all tables.
Conclusion
Comment 6: A Conclusion section is missing.
Response 6: We added a separate conclusion section. Please see L465.
Round 2
Reviewer 1 Report
Comments and Suggestions for Authors
Thanks for responding to all my comments. Here still a few to tidy up the manuscript.
I suggest to change throughout the manuscript the “,” in values with “.”.
Note inconsistency in the list of references and I suggest to check author guidelines for the correct format.
Tables 2 to 5 and 7. Please ensure that for each parameter separately, the mean and SD value are expressed with the same number of decimal places.
Table 6. 2 in O2 needs to be subscript.
Author Response
Thank you for the additional comments and suggestions. All changes are marked in red.
In the final revision, we have noted a typing error. ANCOVA for METs is 0.049, not 0.49 as originally written. We have, therefore, made appropriate changes in the abstract, results section, and Table 4.
Comment 1: I suggest to change throughout the manuscript the “,” in values with “.”
Response 1: Corrected.
Comment 2: Note inconsistency in the list of references and I suggest to check author guidelines for the correct format.
Response 2: Thank you for pointing this out. It is stated in the author guidelines: “Your references may be in any style, provided that you use the consistent formatting throughout. It is essential to include author(s) name(s), journal or book title, article or chapter title (where required), year of publication, volume and issue (where appropriate) and pagination. DOI numbers (Digital Object Identifier) are not mandatory but highly encouraged...“
We have chosen the NLM citation format. After reviewing the reference list, we have corrected the ones marked in red.
Comment 3: Tables 2 to 5 and 7. Please ensure that for each parameter separately, the mean and SD value are expressed with the same number of decimal places.
Response 3: Corrected.
Comment 4: Table 6. 2 in O2 needs to be subscript.
Response 4: Corrected. Corrections have been made throughout the text regarding these parameters (subscript and superscript).
Reviewer 2 Report
Comments and Suggestions for Authors
Authors answered only partially to my comments, for example I can not see how 5kg of body weight and muscle mass can not be statistically significant, and even if it is it must be ssen as considerable difference
Comments on the Quality of English Languageit needs a final revision
Author Response
Thank you for the additional comments and suggestions. All changes are marked in red.
In the final revision, we have noted a typing error. ANCOVA for METs is 0.049, not 0.49 as originally written. We have, therefore, made appropriate changes in the abstract, results section, and Table 4.
Comment 1: Authors answered only partially to my comments, for example I can not see how 5kg of body weight and muscle mass can not be statistically significant, and even if it is it must be ssen as considerable difference
Response 1: Thank you for pointing this out. Even though the differences are not statistically significant, that can be a consequence of a small sample size. We have noted this in the Limitations section. Please see L477.
Thank you for your general comments. If you need any additional clarification, please let us know.
Comment 2: Comments on the Quality of English Language
it needs a final revision
Response 2: Thank you for your comment. In the first round, you were satisfied with the quality of the English language, which was supported by the other two reviewers. Prior to sending the first version, we used a writing assistant software tool as a double-check.